

# The role of coral colony health state in the recovery of lesions

Claudia P. Ruiz-Diaz[1,2], Carlos Toledo-Hernandez[2], Alex E. Mercado-Molina[2,3], María-Eglée Pérez[4,5] and Alberto M. Sabat[3]

[1] Environmental Sciences, University of Puerto Rico, San Juan, Puerto Rico
[2] Sociedad Ambiente Marino SAM, San Juan, Puerto Rico
[3] Department of Biology, University of Puerto Rico, San Juan, Puerto Rico
[4] Department of Mathematics, University of Puerto Rico, San Juan, Puerto Rico
[5] Center for Tropical Ecology and Conservation, CATEC, Río Piedras Campus, University of Puerto Rico, San Juan, Puerto Rico

## ABSTRACT

Coral disease literature has focused, for the most part, on the etiology of the more than 35 coral afflictions currently described. Much less understood are the factors that underpin the capacity of corals to regenerate lesions, including the role of colony health. This lack of knowledge with respect to the factors that influence tissue regeneration significantly limits our understanding of the impact of diseases at the colony, population, and community level. In this study, we experimentally compared tissue regeneration capacity of diseased versus healthy fragments of *Gorgonia ventalina* colonies at 5 m and 12 m of depth. We found that the initial health state of colonies (i.e., diseased or healthy) had a significant effect on tissue regeneration (healing). All healthy fragments exhibited full recovery regardless of depth treatment, while diseased fragments did not. Our results suggest that being diseased or healthy has a significant effect on the capacity of a sea fan colony to repair tissue, but that environmental factors associated with changes in depth, such as temperature and light, do not. We conclude that disease doesn't just compromise vital functions such as growth and reproduction in corals but also compromises their capacity to regenerate tissue and heal lesions.

## INTRODUCTION

Most of the present-day coral reef habitats no longer exhibit the complex community structure that was commonly observed several decades ago. This is particularly evident in the Caribbean where the most important reef species such as the coral-building Caribbean *Acropora palmata*, *A. cervicornis* and the *Orbicella* complex (formerly *Montastraea*), and the predatory reef fish and herbivores such as the black sea urchins and sea fan corals, have dramatically decreased in abundance (*Kim & Harvell, 2002*). These losses have not just changed the seascape of the reefs, but have also caused important ecological alterations to coral survival, growth and reproductive schedules at local and regional

Corresponding author
Claudia P. Ruiz-Diaz, claudiapatricia-ruiz@gmail.com

scales (*Sutherland, Porter & Torres, 2004*; *Hoegh-Guldberg et al., 2007*; *Weil, Cróquer & Urreiztieta, 2009*; *Burns & Takabayashi, 2011*; *Ruiz-Diaz et al., 2013*).

Of the myriad of stressors affecting the viability of corals, disease is currently ranked at the top of the list. Coral diseases are typically diagnosed based on changes in the normal coloration of corals and by the appearance of lesions (partial tissue mortality). Under severe circumstances, such as when a pathogen is highly virulent or the coral host is immune-suppressed, disease-induced lesions can increase in size quickly, killing the colony. However, given a strong immune response, diseased-induced wounds can be contained and either persist for a prolonged period (if the colony is able to contain the disease but not regenerate new tissue) or are temporary (if the colony is able to regenerate tissue over the whole lesion) (*Ruiz-Diaz et al., 2013*).

Several studies have identified wound characteristic as a major factor affecting the rate at which a colony can regenerate new tissue and eliminate a lesion. For instance, several studies agree that regeneration rate decreases with an increase in lesion size (*Bak & Steward-Van, 1980*; *Oren et al., 2001*; *Kramrsky-Winter & Loya, 2000*). Other studies suggest that the area/perimeter ratio of a lesion largely governs the rate of wound healing process (*Lirman, 2000*). Further studies suggest that wound position within the colony (i.e., lesions at the edge of the colony vs. lesion at the center of the colony) determine the wound healing process (*Meesters, Bos & Gast, 1992*).

Many researchers have also linked the ongoing environmental degradation experienced by most coral reefs with the advent of coral diseases, which currently is one of the main sources of lesions on corals. For instance, in a study by *Toledo-Hernández, Sabat & Zuluaga-Montero (2007)*, the capacity of corals to recover from diseases (i.e., lesion recovery) was correlated with turbidity and/or sedimentation. Corals in areas with high turbidity and sedimentation had higher frequencies of disease-induced lesions and larger lesions compared to corals in less degraded habitats. Higher water temperature has been linked to the progression of lesions caused by black band disease, which affects several coral species in the Caribbean and the Great Barrier Reef (*Kuta & Richarson, 2002*; *Haapkylä et al., 2011*). Similarly, nutrient enrichment increased the severity of aspergillosis of *Gorgonia ventalina* and yellow band disease on *Orbicella annularis* and *O. franksi* (*Bruno et al., 2003*). *Muller & Woesik (2009)* showed that white-plague lesion significantly decreased on *Corpophyllia natans* shaded from solar radiation when compared to *C. natans* without shading. Although results from these studies have been useful in advancing our understanding of the healing process on corals, we still lack comprehensive knowledge of how other factors such as the health state of a colony baring lesion, affect the healing process. However, progress has been made. For instance, *Fine, Oren & Loya (2002)* (working with bleached scleractinian corals) and *Ruiz-Diaz et al. (in press)* (working with diseased gorgonians) have shown that diseased corals regenerate man-made lesions slower than man-made lesion inflicted on healthy-looking corals.

Initiatives to mitigate the effects of coral disease lack information about factors affecting the recovery of corals from disease-induced lesions. While we do have some understanding about the factors that make a coral vulnerable to disease (i.e., abnormally high temperature and sedimentation among others) we lack understanding regarding

how the health condition of the coral affects its recovery. The objective of this study is to experimentally test if the health state and variability in environmental factors correlated with depth, significantly influence lesion regeneration on the sea fan *G. ventalina*. To do this, we established eight nursery lines at two depths within the same reef (four nursery lines per depth, 5 m and 12 m). Each nursery line consisted of four fragments from two healthy and two diseased G. *ventalina* colonies. We scraped tissue from some of the healthy fragments and scraped the diseased area of the diseased fragments and followed their recovery through time. Concomitantly, we measured the temperature and light intensity at both depths (5 m and 12 m) to document differences in these factors between depths. We hypothesized that fragments from healthy colonies would regenerate new tissue at a faster rate than those from diseased colonies because, at the start of the experiment, diseased colonies are expected to have an activated immune response and thus fewer resources to allocate to tissue regeneration than healthy ones. We also reasoned that, independent of health state, tissue recovery rate at 12 m would be slower than at 5 m due to reduced light availability.

## METHODS

### Study site

The experiment was conducted in Cayo Largo reef (CL) from April to August 2013. CL is located 6.5 km off the northeastern coast of Puerto Rico (N18°19.09′42″W65°35.01′75″). CL is a patch reef with a coral assemblage dominated by large colonies of *Gorgonia ventalina, Pseudopterogorgia acerosa* and small colonies of the *Orbicella annularis, Acropora palmata* and *Porites astreoides* (for further description of the study area, see *Hernández-Delgado et al. (2006)*). The tissue samples were collected under permit 2012-IC-086 issued to Claudia P. Ruiz Diaz, University of Puerto Rico (UPR) Rio Piedras campus, given by the Puerto Rico Department of Natural Resources, Commonwealth of Puerto Rico.

### Experimental design
#### Nursery lines
A total of eight nursery lines, each of 2.7 m in length and 1 m above the bottom, were established at two depths: 5 m and 12 m (hereafter shallow and deep zones, respectively) at CL (Fig. 1). Four of these nursery lines were established at the shallow zone and the remaining four at the deep zone. To setup the nursery lines, we collected tissue fragments from 16 sea fan colonies (fragment donor colonies) inhabiting an area of about 800 m$^2$ and at depths between 1–1.5 m. Given that sea fans do not exhibit asexual reproduction, selected colonies are assumed to be genetically distinct from each other. Eight of the fragment donor colonies were diagnosed as healthy i.e., fans showing no visual sign of disease or tissue purpling; the remaining eight donor colonies were diagnosed as diseased i.e., fans showing an area colonized by fouling organisms, mainly algae, with a purple tissue ring surrounding the over grown (Fig. 2). Once collected, each health fragment was split in two identical halves of approximately 165.5 cm$^2$, one of which was placed on a shallow nursery line and the other on a deep nursery line. Diseased fragments were split so that the lesion represented approximately 16% of the total surface area of each fragment. Once

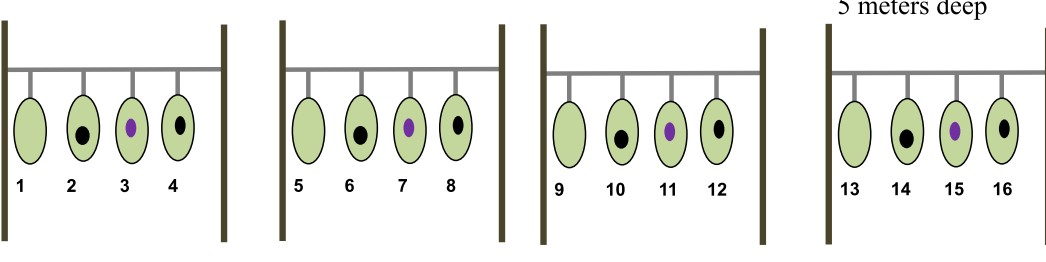

5 meters deep

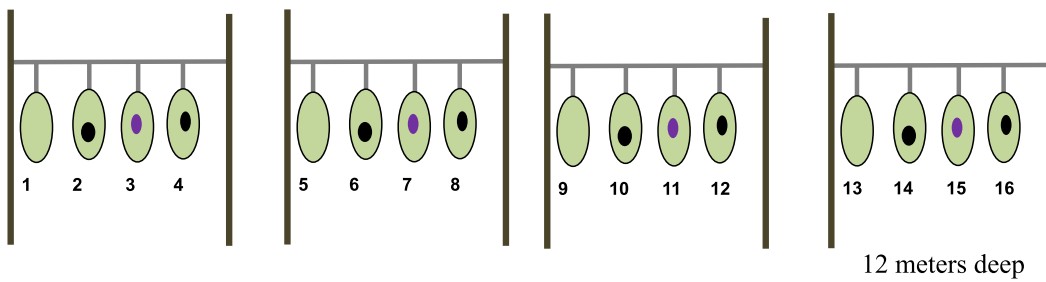

12 meters deep

**Figure 1 Nursery line of the *Gorgonia ventalina* fragments with treatment enumerated.** 1, 5, 9, 13, are healthy fragments (HF). Numbers 2, 6, 10 and 14 are scrapped healthy fragments (HFS). Numbers 3, 7 and 15 are diseased fragments (DF). Numbers 4, 8, 12 and 16 are scrapped diseased fragments (DFS). Light green represents healthy tissue, black oval represents exposed skeleton from experimental scraping, and violet oval represents lesion.

split, one half-fragment from the same donor colony was placed at a shallow nursery lines and the other half at a deep nursery line. Once fully assembled, each nursery line consisted of four colony fragments (two healthy and two diseased) separated by 30 cm each (Fig. 1). Note that no two fragments from the same colony were placed in the same nursery line nor at the same depth.

### Tissue scraping
Tissue scraping was performed to measure the capacity of fragments to regenerate tissue under contrasting health states and environmental conditions. We scraped tissue from one of the healthy and diseased fragments per nursery line, per depth (hereafter HFS and DFS, respectively) (Fig. 1). In the case of HFS fragments, the equivalent of ten percent of the total surface area was scraped from the center of the fragment. For DFS fragments, the total injured area (the area overgrown by fouling organisms plus the purpled tissue) was scraped. Scraping was performed using a metal bristle brush and resulted in the exposure of the axial skeleton in both cases. The remaining healthy and diseased fragments, (hereafter HF and DF respectively), were not subjected to any tissue scraping (Fig. 1). HF fragments were used as sentinels. Tissue mortality in these fragments would signal either an adverse effect of fragmentation or too harsh environmental conditions both of which would invalidate

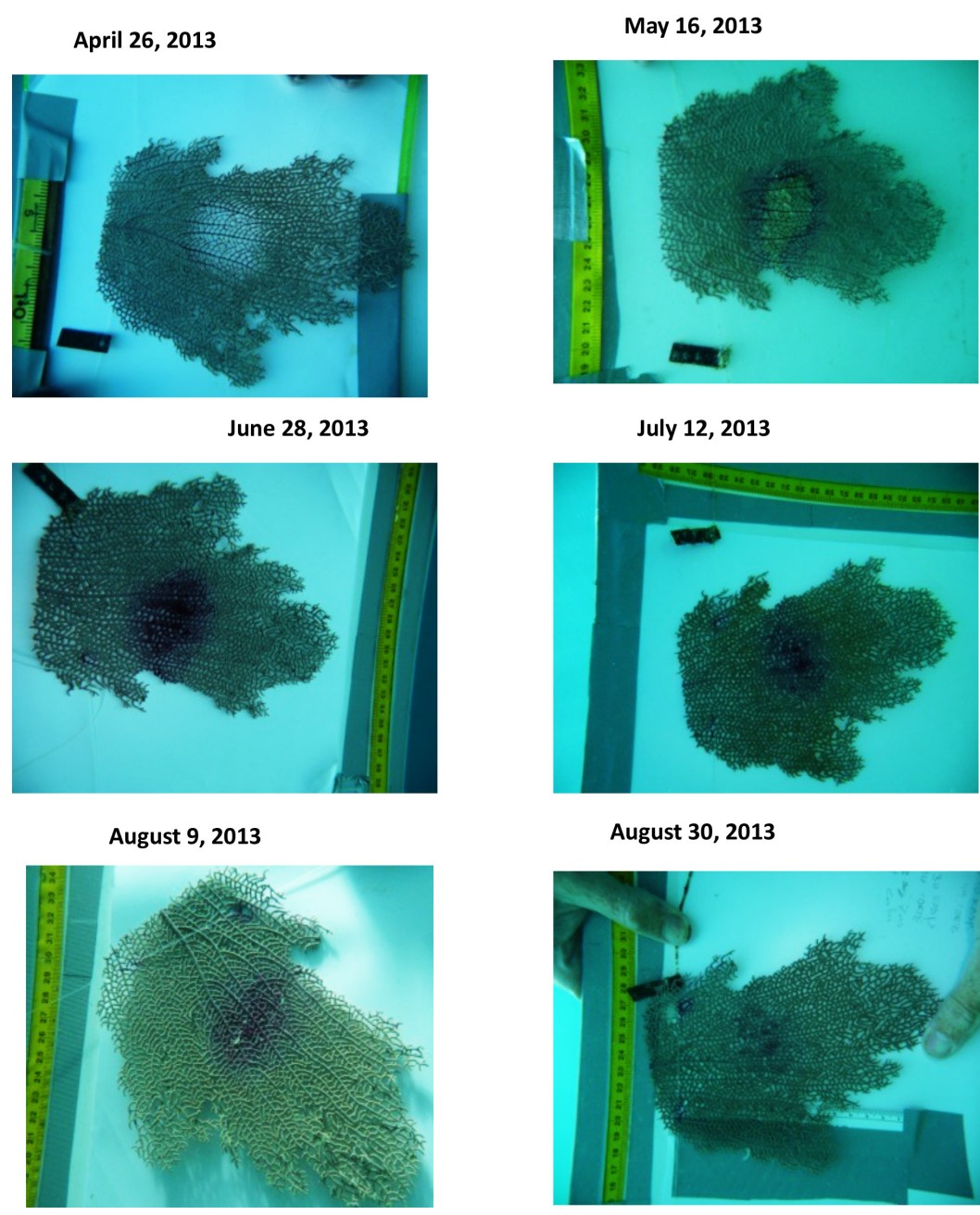

**April 26, 2013**

**May 16, 2013**

**June 28, 2013**

**July 12, 2013**

**August 9, 2013**

**August 30, 2013**

**Figure 2** **Example of wound-healing process.** Close-up pictures of scraped healthy individuals showing the healing process over the course of the experiment.

the experiment. HFS and DFS fragments were included to address the main objective of the study, which is to test the effect of health state on tissue generation. DF are disease fragments with filamentous algae or other fouling growing in the expose skeleton. They were added to the experiment to measure the "natural" regeneration rate of tissue growing over skeleton covered by fouling organisms or/and pathogen(s).

### Tissue regeneration estimates

To document the progression of the wound-healing process, close-up pictures of each fragment were taken every two weeks between April and August 2013 or until lesions healed completely. Lesions were deemed healed (fully recovered), if the bare skeleton was completely covered by healthy tissue. The percent area of the lesion that did recovered at the end of the experiment was estimated by subtracting the area without soft tissue measured at the end of the experiment to the area (bared axial skeleton) measured at the beginning of the experiment, just after scraping the lesion. Image analysis software (Sigma Scan Pro Image Analysis version 5.0 Software) was used to measure all individual and clone fragments. These measurements were validated using *in situ* measurements.

### Environmental variables

To quantitatively determine if environmental conditions differed at each depth (5 m and 12 m), we measured the water temperature and light intensity. Temperature and light were measured using one Hobo Pendant temperature/light data logger 64k-UA-002-64 (Onset Company) at each depth. Data loggers were secured in place using metal rods and a zip tie. Temperature measurements were recorded every 15 min for 14 days from April 26 to May 3, May 16 to June 7, June 28 to July 12, and August 9 to August 23, 2013. Light intensity data was obtained only during the first 10 days after the loggers were placed, as seaweeds typically colonize the logger and affect the readings (C Ruiz-Diaz, pers. obs., 2013).

## Statistical analysis

Lesion recovery was expressed as the rate at which tissue regenerated (in cm$^2$) through time. This can be represented as the slope of a linear regression with time (in days) in the $x$-axis and lesion area in the $y$-axis (log transformed) (*Meesters, Bos & Gast, 1992*). To determine whether depth (5 m and 12 m) and fragment treatments (DF DFS, and HFS) had an effect on the tissue regeneration through time, the slope of each fragment was analyzed using a repeated measure ANOVA, as fragments from the same colony (placed at the shallow and deep nursery lines) are not independent from each other. Statistical analyses were performed using R version 3.1 (R Core Team 2014).

## RESULTS

### Environmental variables and recovery

Light intensity and temperature showed statistical differences between depths (see Table 1). Average temperature at 5 m was 28.555 ±0.012 °C (mean ± SE), while at 12 m it was 28.334 ±0.006 °C. Average light intensity at 5 m was 11203.55 ±459.410Lux, while at 12 m it was 3429.36 ±129.11Lux.

### Tissue recovery

All the healthy sentinel fragments (HF) survived the experiment without any necrosis; in fact, fragments increased in size at both depths. The results from the repeated measure ANOVA analysis performed showed that tissue recovery was only affected by fragment's health state ($F_{2,15} = 5.477$, $p = 0.0317$). Depth ($F_{1,15} = 3.587$, $p = 0.095$) and the interaction between depth and health state showed no significant differences ($F_{5,15} = 3.915$, $p = 0.065$;

**Table 1** *t*-test statistics for light intensity and temperature for different time periods for both the shallow and deep sites. The experimental period lasted between April 26 to August 23, 2014.

|  | April 26–May 3 | May 16–June 7 | June 28–July 12 | August 9–August 23 |
|---|---|---|---|---|
| Light intensity | $t = 15.13$ $df = 363.40$ $p < 0.001$ | $t = 15.52$ $df = 992.63$ $p < 0.001$ | $t = 17.58$ $df = 902.61$ $p < 0.01$ | $t = 17.53$ $df = 897.81$ $p < 0.001$ |
| Temperature | $t = 10.42$ $df = 838.22$ $p < 0.001$ | $t = 17.50$ $df = 3541.62$ $p < 0.001$ | $t = 12.87$ $df = 2274.34$ $p < 0.01$ | $t = 26.72$ $df = 3051.96$ $p < 0.001$ |

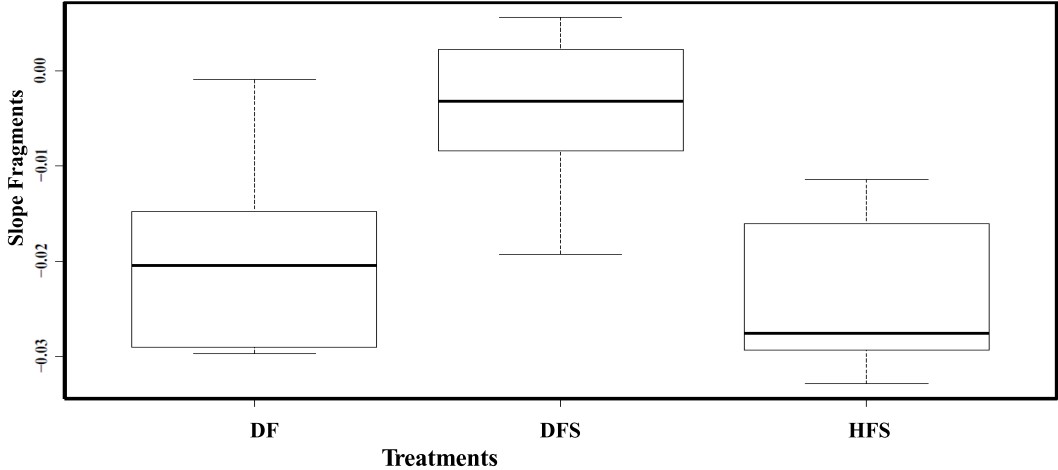

**Figure 3** **Boxplots showing the slopes (rate at which tissue regenerated through time) between health state treatments (healthy and diseased) and fragments.** The boxplot median is represented by the bold line, the extremes of the box are the 1st and 3rd quartile and the whiskers are the maximum and minimum. DF, diseased fragments; DFS, diseased fragments scrapped; HFS, healthy fragments.

Fig. 3B). The results of the Tukey HSD analysis showed significant differences between DFS and HFS (diff = 0.020, $p = 0.001$) and DFS and DF (diff = 0.015, $p = 0.016$).

## DISCUSSION

Coral colonies are very vulnerable to tissue loss due to predation, pathogens, and abrasion, among other factors. Failure to regenerate lost tissue could impair their survivorship by allowing potentially harmful organisms to settle in the exposed skeleton, further infecting healthy areas of the corals. Repair failure could also affect other vital function of corals such the heterotrophic feeding and ultimately growth, in addition to reproduction, as loss of polyps will negatively affect such activities. Thus, tissue regeneration should be of utmost importance in order for coral colonies to reduce the risk of diseases, thereby improving their survivorship, competitive capacity and ultimately reproduction and somatic growth.

Numerous researchers have studied the link between environmental factors, and the frequency and severity of coral diseases. In fact, some of these studies have argued that as climate change continues to exacerbate these factors, so will be the physiological stress associated with it, and that consequently the frequency and severity of coral disease, will also

increase (*Kuta & Richarson, 2002*; *Haapkylä et al., 2011*; *Cróquer et al., 2006*; *Williams et al., 2014*). In comparison, studies addressing how the health state of corals affects the coral's capacity to repair are by far less common (however, see *Mascarelli & Bunkley-William, 1999*; *Fine, Oren & Loya, 2002*; *Ruiz-Diaz et al., in press*). This study is an attempt to address this knowledge gap by documenting the relationship between the recovery dynamics of healthy and diseased coral colonies and environmental factors such as temperature, light intensity while controlling for genetic variability.

## Effect of the state of coral health on lesion recovery

This study shows that the health state of colonies (i.e., being diseased or healthy) has a significant effect on the tissue repair capacity of sea fans. All healthy fragments, regardless of the depth where they were placed (thus regardless of temperature and light regimes), exhibited full recovery whereas diseased fragments did not. Furthermore, scraped healthy fragments healed faster than scraped diseased fragments (i.e., on average 78 days vs. 97 days, respectively). It is possible that genetic differences among colonies, which may have lead to different levels of susceptibility to disease in the first place, might have lead to the observed differences in healing rate. However, the result is that unscraped diseased fragments (DF) healed at a significantly slower rate than scraped ones (DFS) supports that tissue with lesion cannot heal as fast as tissue without a lesion even if they come from the same colony. In other words, growing tissue over a skeleton covered with fouling organisms is a slower process because it is more costly, as the coral is competing for space and also allocating resources into tissue regeneration. By contrast, scraped fragments can allocate resources into tissue regeneration.

The results of the experiment agree with our initial hypothesis, which stated that the health state does affect the capacity of fragments to recover. In fact, our results show that being diseased negatively affects the capacity of fragments to recover. These results also concur with several authors who have argued that the diseased condition negatively affects the tissue regeneration capacity of corals. For instance, *Mascarelli & Bunkley-William (1999)* compared the rates of tissue regeneration of *Orbicela annularis* corals under contrasting health conditions (healthy and artificially bleached fragments) and reported that healthy ramets did not just heal completely but also recovered faster than diseased ones. By contrast, two of the bleached ramets died, and the remaining fragments did not exhibit full recovery. Likewise, *Ruiz-Diaz et al. (in press)* scraped naturally occurring lesions from sea fan colonies and as control scraped the equivalent of 10% of the surface area of healthy sea fan colonies, and found that tissue recovery was significantly slower in diseased fans when compared to healthy fans. A plausible explanation for these differences is that diseased colonies have fewer resources to invest into tissue repair as their resources were already compromised by the immune response prior to scraping (*Nagelkerken et al., 1997*). Further evidence in support of this explanation of resource limitation would have been obtained by contrasting regeneration rates of healthy fragments from diseased colonies with that of diseased and healthy fragments from healthy colonies; however, we did not included healthy fragments from diseased colonies in our experimental design. Corals, like all living organisms, have finite resources to allocate into several vital functions

such as growth, reproduction, immune defense or lesion regeneration. Given these resource constraints, the allocation of resources into certain vital functions, such as immune defense, means that fewer resources could be available for lesion regeneration (*Oren et al., 2001*). Several studies conducted on a variety of corals support this hypothesis. For instance, *Petes et al. (2003)* working on sea fan coral *G. ventalina* reported reproductive suppression in diseased colonies, presumably due to a shift in resource allocation from reproduction to immunity. Similarly, *Palmer, Bythell & Willis (2010)* suggest that *Porites* sp. invests considerably more resources into immune constituents such as melanin biosynthesis than *A. millepora*. This investment of resources into immunity provides *Porites* with a higher disease and bleaching resistance. By contrast, *A. millepora* invests more resources into growth compared to *Porites*, although at a cost in reduced immunity, as acroporids are among the corals most susceptible to bleaching and disease.

### Effect of depth on lesion recovery

One of the main concerns of the scientific community is that changes in environmental conditions could induce physiological stress on corals (*Alker et al., 2004*). These stresses could impair vital life history traits such as grow, reproduction or even the capacity of corals to recover after a disturbance. In our study, however, environmental factors associated with changes in depth, showed no evident effects on the capacity of sea fan fragments to regenerate tissue, even though, the parameters measured were statistically different between depths. Our failure to detect depth effects could have several explanations, not necessarily mutually exclusive. For instance, it could be possible that the difference in environmental factors recorded between 5 m and 12 m were not sufficient to induce physiological stresses on the fragments, thereby not precluding their capacity to regenerate tissue. Alternatively, it could be that there was a depth effect but it manifested on other life history traits such as reproduction or somatic growth, in which case we were not able to detect it. It is also plausible to argue that sea fans are rather tolerant to changes in environmental conditions. Indeed, *Ruiz-Diaz et al. (in press)* found no differences in tissue recovery of in *G. ventalina* inhabiting reefs with contrasting water quality.

## CONCLUSIONS

Diseases of corals not just compromise vital functions such as growth and reproduction, but also compromise their recovery capacity. Arguably, resources invested against pathogens could also be the same driving the tissue repair as stated by limited budget theory proposed by *Oren et al. (2001)*. This raises questions regarding the sharing of resources and resource depletion. For instance, in the eventuality of two simultaneous but different immunological insults, how corals should prioritize its resources to respond to both events? How intense should a disturbance be in order to induce immune responses that affect several life history traits? It that regard, it could be possible that the environmental conditions in this study may have indeed caused stress on the sea fan fragments, but these stresses were manifested in other vital functions such as reproduction or growth which were not studied in this work. Our study also shows that sea fans are very robust corals which can tolerate variable

environmental conditions; this may explain why this species thrives relatively well in many coral reefs across Puerto Rico regardless of environmental degradation.

## ACKNOWLEDGEMENTS

Ruber Rodríguez and Francisco J. Soto for field assistance, and Paul Furumo, Molly Ramsey, Cheryl Woodley and Misaki Takabayashi for their critical reviews.

### Funding

This study was supported in part by institutional funds of the UPR-RP, UPR Sea Grant (NOAA award NA10OAR41700062, project R-92-1-10) and UPR-Sea Grant (Seedmoney) to C.P.R-D and the Puerto Rico Center for Environmental Neuroscience (NSF grant HRD #1137725). The funders had no role in study design, data collection and analysis, decision to publish, or preparation of the manuscript.

### Grant Disclosures

The following grant information was disclosed by the authors:
UPR-RP, UPR Sea Grant: NA10OAR41700062.
UPR-Sea Grant (Seedmoney) to C.P.R-D and Puerto Rico Center for Environmental Neuroscience: #1137725.

### Competing Interests

The authors declare there are no competing interests.

### Author Contributions

- Claudia P. Ruiz-Diaz performed the experiments, analyzed the data, contributed reagents/materials/analysis tools, wrote the paper, prepared figures and/or tables, reviewed drafts of the paper, guided all the work of the manuscript.
- Carlos Toledo-Hernandez and Alberto M. Sabat conceived and designed the experiments, performed the experiments, analyzed the data, contributed reagents/materials/analysis tools, wrote the paper, reviewed drafts of the paper.
- Alex E. Mercado-Molina conceived and designed the experiments, performed the experiments, wrote the paper, reviewed drafts of the paper.
- María-Eglée Pérez performed the experiments, contributed reagents/materials/analysis tools, reviewed drafts of the paper.

### Field Study Permissions

The following information was supplied relating to field study approvals (i.e., approving body and any reference numbers):

The tissue samples were collected under permit 2012-IC-086 issued to Claudia P. Ruiz Diaz, University of Puerto Rico (UPR) Rio Piedras campus, given by the Puerto Rico Department of Natural Resources, Common Wealth of Puerto Rico.

## Data Availability

The research in this article did not generate any raw data.

## Supplemental Information

Supplemental information for this article can be found online at http://dx.doi.org/10.7717/peerj.1531#supplemental-information.

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
