# Peer review of "The role of coral colony health state in the recovery of lesions"

_PeerJ, doi:10.7717/peerj.1531_

## Round 0.1 · original submission · Major Revisions

I am sorry for the delay, but I have been waiting on a third review that does not seem to be coming after all. Rather than delay you any longer, I will proceed with the two reviews that I have in hand. While both are enthusiastic about the question and generally positive about the work, they also have substantial suggestions for improvement. Even the more positive referee states that they don’t feel that the evidence presented is sufficiently strong to support conclusions drawn. The more critical referee is unable to evaluate the study because of the unanswered questions about the experimental design. I consider both serious criticisms of the manuscript, and cannot accept the paper until these criticisms are addressed. As such, I am asking for a major revision of the manuscript to address the comments of these referees so that they are able to evaluate your manuscript, and will try to use the same reviewers with your revised submission.

Reviewer 1 ·

Basic reporting

Ruiz-Diaz et al. addressed a very interesting question of how original disease state of a gorgonian colony and its habitat affect its ability to recover from tissue-damage. This is a simple, yet important question that is not well investigated. The manuscript, as it is written, lacks in clarity especially for the experimental design. Therefore, I was unable to evaluate its publication merit completely. I believe it is a publishable paper and encourage the authors to revise and resubmit.

Conclusion section is rather awkward. Authors introduce new thoughts in the Conclusions without much context. For example the sentence in lines 301-304 seems to be a possible explanation as to why the results countered the null hypothesis, which was not included in the Discussion. The entire conclusion section should be re-written to succinctly summarize the study and state implications of the findings in a larger context.

Justify why tissue loss/scraping was used as a condition to subject gorgonians to. Is tissue loss a condition that these gorgonians are subjected to naturally by predation or wave exposure that you intended to simulate? Or is it simply a “non-healthy” state that could be experimentally imposed? I understand that the authors have successfully used scraping as a potential method to remove a lesion and allow recovery in gorgonians.

Minor comments
Consistently use “differential environmental factors associated with depth” rather than switching between “environmental factors” and “depth”.

Ln 80: change “abnormal” to “abnormally”
Ln 97-99: cite papers for this statement (Ruiz-Diaz et al. 2015?)
Ln 110: change to “Commonwealth”
Ln 118: the term “over grown” is unclear here. Do you mean “lesion”?
Ln 155: replace “seaweed” with “algal”
Ln 197: “ramet treatments” is unclear. Do you mean “among ramets from the same genet”?
Ln 204: replace “is it” with “it is”
Ln 225: replace “impair” with “make a difference in”
Ln 254: replace “it’s” with “it is”
Ln 278: replace “didn’t” with “did not”

Experimental design

Description of experimental design unclear
– Modify Fig 1 so that it pictorially shows the flow of what happened to each fragment (e.g. Ramet #1 - healthy genet #1 scraped; Ramet #2 - healthy genet #2 not scraped)
– Use of the terms ramet and genet is confusing. Although these terms are defined early on, they are used to describe particular fragments, for example in Figure 1. If you use them as the above example to describe the fragments, it would be easy to understand the experimental design. Perhaps “ramets” could be replaced in some instances by “fragments”?
– What are blocks? What is the relevance of including the area of four genets included in a block (Ln 114-115)?

As far as I could tell, there were no fragments (ramets) from the healthy part of diseased colonies (genets). Is this correct? If this is the case, the possibility that colonies that had succumbed to the disease were somehow immunologically or physiologically weaker than the healthy colonies cannot be ruled out as an explanation to the observation. Although it would improve to include fragments from the healthy part of diseased colonies as another ramet in the future studies, including this possibility in the Discussion should suffice in this paper.

At what depth were all colonies collected from originally? Inevitably, some fragments would have been translocated to a depth that was different from their original habitat. Please discuss how this could have affected the results.

Validity of the findings

I am happy to re-review the findings after my questions for the experimental designs are answered, and I have a clearer understanding of what happened in this experiment.

·

Basic reporting

While I agree with the basic conclusion of the authors: that health condition has a substantial effect on healing efficiency, I don’t think that the evidence presented is sufficiently strong to support conclusions related to some of the environmental factors measured in the study.

Experimental design

1. The authors claim to be using different genets, but provide no genetic evidence to support this claim. Please clarify/justify the use of genet vs colony.

2. Please clarify the experimental design description regarding the evaluation of depth effect. Were fragments from each of the four colonies reciprocally transplanted as well as placed at the same depth as the parent colony? Meaning 4 shallow fragments were place in each of the 4 deep plots, 4 shallow fragments into 4 shallow plots, 4 deep fragments into 4 shallow blocks and 4 deep frags into 4 deep plots? I believe this was the design, but Fig 1 doesn’t convey a reciprocal transplant.

3. Though light intensity (lux) is an important factor in coral growth and development, type of light (spectra; PAR) is equally important. The authors are using light intensity data to support an explanation based on photosynthetic responses, which also requires certain wavelengths of light (PAR). The authors should review their discussion related to light effects, and revise to either limit their discussion to effects of light intensity or provide further discussion that explains why lux measurements are relevant to discussing photosynthetic effects.

4. Though water motion is important for coral growth and I applaud the authors for attempting to include this variable in their studies, I am not convinced from the information provided that the design is supportive of their conclusions. The measurements were taken during a 2.5 day period 4-5 months after the field study. The authors provide no evidence that the study area maintains constant water motion throughout the year or why these measurements represent conditions during their healing studies. I recommend removing this test variable or provide a rational for its inclusion.

Validity of the findings

The authors provide convincing data that healthy G. ventalina heal faster than diseases and that depths used in this study did not affect the healing rates. I am not convinced that the temperature differential is biologically relevant, even though the authors indicate a statistical difference between depths. I would anticipate light would not have an effect on colonies acclimated to a given depth. Although the reader has to infer there were not effects of light on the reciprocally transplanted ramets, it is interesting that deep corals moved to higher intensity light showed no effect. The authors may want to expand their discussion (and results) and speculate as to why light intensity showed no adverse effects on healthy or diseased ramets moved from deep to shallow.

I am not convinced that the authors are using the appropriate data to conclude water motion had no effect, despite finding differences between deep and shallow sites. Without having measurements during the study period the authors cannot say with certainty that water motions differed between depths. I recommend the authors revise this portion of their manuscript.

Additional comments

Detailed Comments:
1. Line 44 – Acropora palmata – change the palmate to palmate

2. Line 45 – Montastrea should be Montastraea

3. Line 64 – i.e., - comma needs to be inserted

4. Line 80 – i.e., - comma needs to be inserted

5. Line 81 – affects

6. Line 89 – ramets should be ramet treatments

7. Lines 95-99 – This sentence seems to be better placed in the discussion or perhaps in the methods. To this point in the introduction there is no context provided for the explanation of scraping a lesion as part of a methodology. I would recommend removing this sentence from the introduction.

8. Line 116 – i.e., is italicized here but not in previous uses, be consistent

9. Line 125 – brushes should be singular: brush

10. Line 135 – half – …each resulting half…

11. Line 136 – area

12. Lines 139 – 140 – genet and ramet

13. Line 147 - …all fragment and ramet pictures.

14. Lines 154 – 162 – Measurements of temperature and light intensity are indicated in 14 day intervals in each of the 4 months of study however lines 154-156 indicate only the first 10 days of placement. This information needs to be clarified. Does this mean the first 10 days out of every 14 day deployment of the data logger? Or the first 10 days, period? Also is the overgrowth algae or actually seaweed?

I am concerned with the short duration and timing of the environmental data for water motion (2.5 days outside of the study period and not even in the same season) that were collected. How is this justified?

15. Lines 164-167 – This seems to be missing a reference. Was the lesion recovery rate based on procedures by others?

16. Lines 189-190 – How does Lux relate to PAR, since the authors are relating Lux to a discussion involving photosynthesis.

17. Lines 190-194 – the measurement of water motion in Dec and January, during the winter vs during the experiment does not seem appropriate and should not be used to make any inference. I would recommend removing this information or provide sufficient justification as to why it is relevant to the study.

18. Please clarify the experimental design as commented above related to reciprocal transplants.

19. Line 204 – remove extra space before period after diseases

20. Line 205 – seems as though part of the sentence is missing - ….continue to exacerbate, the …. This seems to beg the question of exacerbate what? Please rephrase.

21. Line 214 – comma after i.e. – also be consistent with italics or not

22. Line 240 – ‘light-levels’ should be singular: light-level regimes

23. Lines 238-243 – there was no mention of measuring photosynthetic efficiency in the methods section, thus I am not sure how the authors can make claims that the 29% reduction in light levels between shallow and deep didn’t “significantly reduce photosynthetic-derived resources.” This in fact may be true due to the adaptation or acclimation of the sea fans at the two different depths, but the statement is not supported by the data provided. This passage needs to be revised or supported with data.

Further the authors indicate they replicated the light intensity in the laboratory, however to make this claim, not only the intensity but the spectra would need to be the same for a true replication. No mention of the spectra used in creating this simulation.

24. Line 248 – remove comma after et al.

25. Lines 251-258 – I am not convinced that the authors can use their water motion measurements that were not taken during the study period and assume they were the same in Dec/Jan vs April-Aug. To use this data, the authors need to provide further evidence that these measurements are representative of these locations throughout the year or are comparable to the water movement velocities in April-Aug during the study period.

26. Lines 270 – 274 – The assumption that scraping ‘diseased lesions’ frees the colony from infections is not supported in theory or data. This assumption does not take into account what disease affliction the coral is experiencing, if it is indeed an infectious agent and ignores the inter-connectivity of tissues that may have sub-clinical (i.e., not observable) disease conditions. In addition the fact that sea fans have a pink/purple reaction does not necessarily mean it is affected with an infectious disease agent, but rather can also develop discoloration from physical trauma. The authors should review this assumption and provide further support or rephrase.

27. Line 294 – remove italics from ‘a’

29. The reference style is inconsistent and needs revision for consistency and accuracy. Some points to consider:
a. Title - some have primary words capitalized others have only the first word capitalized
b. Genus/species is not italicized in all instances, also in some instances both are capitalized, this should be corrected (e.g., Oren et al)
c. PLoS one – The new form is all caps
d. There is extra punctuation in several instances
e. Usually in the title, the first word after a colon is capitalized (ex Frade et al)
f. Incomplete journal name (e.g., Kuta – should be Coral Reefs)
g. Title of books are usually capitalized
h. Journal titles should have each primary word capitalized and italicized (e.g., Sebens et al; Ruiz-Diaz et al 2013)

30. Table 1 – it would be helpful to include the measurements for light and temperature along with the statistics.

31. Figure 1 – if a reciprocal transplant of ramets was conducted, the authors may want to clarify this in the legend and diagram. Alternatively a second diagram indicating the ramet design may be appropriate, if indeed a reciprocal transplant of deep ramets to shallow and vice versa infact did occur.

31. Figure 2 – It would be helpful to white balance the photos to provide better clarity.

32. Figure 3 – It would be helpful to include a summary statement of the statistics in the figure legend.

---

## Round 0.2 · Major Revisions

Referee 2 was satisfied, but Referee 1 was not and more importantly is still unable to decipher your experimental design. I know this will not be what you wanted to hear, but given the fact that the referee is unable to figure out your design, it will be difficult for others to do the same which severely limits the utility and repeatability of the work. If the referee is interpreting your design correctly, they feel that the analyses need to be re-run to adjust for the experimental design. If they are not interpreting your design correctly, then the text is not clear enough for the reviewer to interpret what you have done. In either case, further revision is needed to clarify the experimental design or justify the statistical analyses. The referee also feels that the issue of genetic predisposition of the colonies to disease has not been sufficiently addressed in the revision, and I am inclined to agree with them.

The referee is willing to waive their anonymity to assist you with the revisions if that would help, and I can put you in direct contact with them if you would value that input?

Reviewer 1 ·

Basic reporting

The authors have made a genuine attempt to respond to reviewers' comments. The manuscript has improved in clarity and simplicity.

There are some typographical errors.
Correct "scrapping" and "crapping" to "scraping".
Remove references for water flow that are no longer cited. (e.g. Sebens et al. papers)
Change "over grown" in line 119 to "lesion"

Experimental design

The description of design has improved in clarity somewhat. I still do not understand the difference between "individual" and "clone". For example, aren't "HI" and "HC" both fragments from colonies? If I am understanding the authors correctly, the hierarchical design is better described as:

Healthy/Diseased >> Colonies >> Fragments ("clones") >> scraping/non-scraping

By analyzing if there is a difference between colonies of the same condition (healthy/diseased), you can test for the effect of inter-colonial differences (what the authors are calling "individual").

The statistical analyses have to be re-run to adjust for this design.

Validity of the findings

I could not find discussion to address the effect of possible genetic predisposition of colonies that could not be ruled out because the experiment did not include the healthy part of diseased colonies (my comment #16 in the original review). Do lines 268-270 address this? If so, it's really indirect and insufficient.

---

## Round 0.3 · Minor Revisions

I understand that you and Misaki have discussed the review offline and resolved the issues, so I am returning a decision of minor revisions so that you can upload the final version of the manuscript. I look forward to seeing it.

Reviewer 1 ·

Basic reporting

I am satisfied with the revisions. I believe that the manuscript is worthy of publication in PeerJ.

Experimental design

The revised experimental design is great.

Validity of the findings

The findings are valid now.

---

## Round 0.4 · accepted · Accept

Thank you for working through the concerns with the referee, and for your revised manuscript. I am happy to accept it now and move it forward into production.